# Sexual norms and the intention to use healthcare services related to female genital cutting: A qualitative study among Somali and Sudanese women in Norway

**Mai Mahgoub Ziyada** [1,2] *, **Inger-Lise Lien** [1], **R. Elise B. Johansen** [1]

**1** Section for Trauma, Catastrophes and Forced Migration—Adults and Elderly, Norwegian Centre for Violence and Traumatic Stress Studies, Oslo, Norway, **2** Institute of Health and Society, Faculty of Medicine, University of Oslo, Oslo, Norway

\* m.m.ziyada@nkvts.no

**Data Availability Statement:** All relevant data are within the manuscript.

**Funding:** ILL and MMZ received funding from the research council of Norway (https://www.

## Abstract

### Background

Female Genital Cutting (FGC) is a traditionally meaningful practice in Africa, the Middle East, and Asia. It is associated with a high risk of long-term physical and psychosexual health problems. Girls and women with FGC-related health problems need specialized healthcare services such as psychosexual counseling, deinfibulation, and clitoral reconstruction. Moreover, the need for psychosexual counseling increases in countries of immigration where FGC is not accepted and possibly stigmatized. In these countries, the practice loses its cultural meaning and girls and women with FGC are more likely to report psychosexual problems. In Norway, a country of immigration, psychosexual counseling is lacking. To decide whether to provide this and/or other services, it is important to explore the intention of the target population to use FGC-related healthcare services. That is as deinfibulation, an already available service, is underutilized. In this article, we explore whether girls and women with FGC intend to use FGC-related healthcare services, regardless of their availability in Norway.

### Methods

We conducted 61 in-depth interviews with 26 Somali and Sudanese participants with FGC in Norway. We then validated our findings in three focus group discussions with additional 17 participants.

### Findings

We found that most of our participants were positive towards psychosexual counseling and would use it if available. We also identified four cultural scenarios with different sets of sexual norms that centered on getting and/or staying married, and which largely influenced the participants' intention to use FGC-related services. These cultural scenarios are the virgin, the passive-, the conditioned active-, and the equal- sexual partner scenarios. Participants

forskningsradet.no/en/) for the project number
262757. REBJ received no specific funding for this
work. The funders had no role in study design, data
collection and analysis, decision to publish, or
preparation of the manuscript.

**Competing interests:** The authors have declared
that no competing interests exist.

with negative attitudes towards the use of almost all of the FGC-related healthcare services
were influenced by a set of norms pertaining to virginity and passive sexual behavior. In con-
trast, participants with positive attitudes towards the use of all of these same services were
influenced by another set of norms pertaining to sexual and gender equality. On the other
hand, participants with positive attitudes towards the use of services that can help to
improve their marital sexual lives, yet negative towards the use of premarital services were
influenced by a third set of norms that combined norms from the two aforementioned sets of
norms.

## Conclusion

The intention to use FGC-related healthcare services varies between and within the different
ethnic groups. Moreover, the same girl or woman can have different attitudes towards the
use of the different FGC-related healthcare services or even towards the same services at
the different stages of her life. These insights could prove valuable for Norwegian and other
policy-makers and healthcare professionals during the planning and/or delivery of FGC-
related healthcare services.

## Introduction

In the last decade, migrant health and migrants' equitable access to healthcare has received
increased attention in Europe [1–4]. This attention is reflected in the significant increase in the
body of literature on the provision and utilization of healthcare services related to Female Gen-
ital Cutting (FGC) in the European context [5–12]. FGC is mostly prevalent in 30 countries in
Africa, Asia and the Middle East [13], but over half a million immigrant women and girls liv-
ing in Europe are estimated to have undergone FGC [14]. In Norway, around 17,300 girls and
women are estimated to have undergone FGC prior to arrival in the country; half of them are
of Somali origin [15].

The World Health Organization (WHO) classifies FGC into four types, where the most
extensive form is type III (infibulation) [16]. Infibulation is characterized by the sealing of the
vulva (leaving a small hole for passing both urine and menstrual blood) through the cutting
and apposition of the labia majora and/or minora, with or without excision of the clitoris [16].
Women and girls who underwent FGC have a higher risk of experiencing a series of long-term
genitourinary, obstetrical, and psychosexual health problems than those who did not [17–19].
The occurrence and severity of these health problems increase with the extensiveness of FGC.
Genitourinary problems include clitoral inclusion cysts, keloids, dysmenorrhea, hematometra,
hematocolpos, urinary tract infections, dribbling, poor urinary flow, and prolonged micturi-
tion [12, 17, 20–22]. The obstetrical problems include increased risk of cesarean section, episi-
otomy, perineal tears, postpartum hemorrhage, and stillbirths [18, 23]. Psychosexual health
problems include symptoms of anxiety, depression, dyspareunia, lack of sexual desire and
reduced sexual satisfaction [19, 24–30]. Nevertheless, there is increasing evidence that the asso-
ciation between FGC and adverse psychosexual health outcomes is context- dependent [12, 27,
31–35]. In contexts where FGC is positively regarded, as it is often the case in FGC high preva-
lent countries, women and girls who underwent FGC are less likely to complain of psychosex-
ual health problems. In contrast, women and girls who underwent FGC and then migrated to
Western countries, where FGC is negatively regarded, are more likely to report psychosexual
problems [27, 31–35].

Therefore, many European countries, including Norway, have established specialized clinics to meet the healthcare needs of immigrant women and girls who underwent FGC and to provide them with appropriate and sensitive healthcare [36]. The FGC-related healthcare services generally provided in these clinics are deinfibulation, removal of keloids and cysts, clitoral reconstruction, and psychosexual counseling.

Deinfibulation is a minor surgical procedure performed before or after marriage (premarital or marital) on infibulated women and girls who underwent FGC to expose both the vaginal and urethral openings [37–40]. It is typically performed for obstetrical reasons (e.g. to enable childbirth), but can also be performed for gynecological and genitourinary reasons (e.g. to ease urinary and/or menstrual flow, enable sexual intercourse and reduce sexual pain). Removal of keloids and cysts is often conducted in conjunction with deinfibulation.

Clitoral reconstruction is a surgical procedure developed to reduce vulvar and clitoral pain, as well as to improve sexual function and body image in women and girls who underwent any type of FGC involving the removal of the clitoris [41–46]. The idea behind clitoral reconstruction is that the clitoral glans is the part of the clitoris that is removed during FGC, while other parts such as the crura and the body are left intact underneath the ensuing scar tissue. Therefore, it is possible to reconstruct/replace the cut part of the clitoris through the removal of the scar tissue, the exposure of the clitoral stump, and the creation of neo-prepuce. Nevertheless, independent evidence on the safety of clitoral reconstruction is scarce [39, 41].

Psychosexual counseling, on the other hand, is deemed both safe and effective in enhancing the body image, self-confidence, and subsequently the sexual function of women and girls who underwent FGC. Therefore, the WHO guidelines on the management of FGC-related health problems recommend psychosexual instead of clitoral reconstruction for the enhancement of sexual function [39, 41, 45].

In Norway, out of the aforementioned FGC-related health services, only deinfibulation and removal of genital cysts and keloids are available [36]. Neither clitoral reconstruction nor psychosexual counseling with special competence on FGC is available. Considering that psychosexual problems associated with FGC are more relevant in countries of migration, the unavailability of such services in Norway could be a cause for major concern. Nevertheless, already available FGC-related services in Norway are underutilized [47], possibly because of sociocultural beliefs pertaining to premarital sexual behavior [9, 48]. Therefore, prior to the development of any new services, it is important to explore whether women and girls in Norway who underwent FGC have a need for such services, and whether they intend to use such services were it to be available. In this article, we aspire to answer the following questions: *What are the attitudes of women and girls, who underwent FGC and who lives in Norway, towards the utilization of FGC-related healthcare services, regardless of their availability in Norway? What are the perceived norms pertaining to these services? Would these women and girls use psychosexual counseling and/or clitoral reconstruction were they to be offered?*

## Theoretical perspectives

The analysis in this article is primarily informed by the theory of planned behavior and to a lesser extent by the theory of sexual script. As clitoral reconstruction and psychosexual counseling are unavailable in Norway, it is unlikely that our participants have ever used these services. By focusing on the "intention" to use rather than the actual use, the theory of planned behavior [49] accommodates the exploration of attitudes and norms related to the utilization of unavailable services.

In the theory of planned behavior [49], the *intention* to perform a certain behavior is postulated to be the most important determinant of that behavior. Also, the person's *intention* to

perform the behavior is determined by that person's *attitude* towards performing the behavior, the *subjective norm* associated with the behavior, and the person's *perceived control* over the performance of the behavior. A person is considered to have a *positive attitude* towards a behavior when that person holds strong beliefs pertaining to the positive value of performing that behavior, and vice versa. Similarly, the *subjective norm* is considered positive when the person believes that important referents such as parents, spouses, etc. expect him/her to perform the behavior and he/she is motivated to meet the expectations of these referents. Still, *perceived control*, the degree to which the person believes he/she can exercise control over the performance of the behavior, is considered the most important determinant for the *intention* to perform the behavior.

In this study, we found that the attitudes and norms pertaining to the intention to utilize deinfibulation, clitoral reconstruction and psychosexual counseling were predominantly related to sexual norms. This led us to the sexual script theory.

In the sexual script theory [50], sexuality is understood as scripts learned from available *cultural scenarios*. These *cultural scenarios* are described as values and principles that are collectively developed and held, and which act as the reference for what to count as sex and sexual situation, as well as for what to consider as acceptable sexual behaviors. In interpersonal relationships, these *cultural scenarios* can be individually adapted as *interpersonal scripts* and possibly internalized as *intrapsychic scripts*.

How these two theories helped to inform our analysis is described in the methods and together with the findings.

## Methods

This article is based on data from a larger explorative qualitative study examining the perceptions and experiences of Norwegian Somali and Sudanese pertaining to their FGC-related healthcare needs and healthcare services. The data was collected in the period 2016–2018 utilizing participant observation and semi-structured repeat interviews with 26 participants.

The study participants were recruited from one of three Norwegian cities (Oslo, Drammen, and Trondheim) that accommodate a high number of girls and women with origins from one or more of the four largest groups in Norway subjected to FGC, and in particular infibulation, i.e. Somali, Sudanese, Eritreans, and Ethiopians. To facilitate between-groups comparisons, we decided to focus on only two groups: 1) the Somali, as they constituted the largest group in Norway subjected to FGC and 2) the Sudanese, because of the similarities between them and the Somali in terms of prevalent types of FGC, religion, and cultural values such as chastity and marriageability [9].

### Participants, recruitment and data collection

The 26 participants we interviewed were between 16–63 years old (see Table 1). Six of these participants were initially recruited as key informants, but their interviews thereafter expanded and also included personal FGC-related experiences. They were purposefully recruited based on their extensive knowledge of Somali and Sudanese immigrants in Norway gained through their work experiences in the educational and healthcare systems, as well as local non-governmental organizations. The remaining 20 participants were recruited through different starting points and with purposeful diversity in a select number of characteristics (see Table 1). We also made special efforts to recruit participants in the age group 16–25 years since they were mostly underrepresented in previous FGC research in Norway.

Potential participants were informed about the aim of the study, the voluntary nature of participation, and further process if they were to agree to participate. They were also oriented

**Table 1. Overview of participants' characteristics.**

| Characteristic | Participants | |
|---|---|---|
| | Other participants (n = 20) | Key informants (n = 6) |
| Background | | Withheld for anonymity |
| Somalia | 9 | |
| Sudan | 11 | |
| Age (years) | | |
| 16–21 | 8 | 1 |
| 22–27 | 7 | 1 |
| 28–33 | 1 | 0 |
| 34–39 | 2 | 0 |
| 40–45 | 1 | 2 |
| 46–51 | 1 | 0 |
| 52–57 | 0 | 1 |
| 58–63 | 0 | 1 |
| Marital status | | |
| Single | 9 | Withheld for anonymity |
| Married | 7 | |
| Divorced | 4 | |
| Have children | | |
| Yes | 8 | Withheld for anonymity |
| No | 12 | |
| Education | | |
| ≤ Middle school | 2 | 0 |
| High school | 8 | 1 |
| College | 7 | 3 |
| Graduate school | 3 | 2 |
| Type of FGC | | |
| Type I | 2 | Withheld for anonymity |
| Type II | 4 | |
| Type III | 14 | |
| Length of stay | | |
| < 1 year | 1 | 0 |
| 1–5 years | 6 | 0 |
| 6–10 years | 5 | 0 |
| >10 years | 8 | 6 |

about the plan for data storage and management. All those who agreed to participate were asked for their consent. According to the Norwegian Health Research Act, the general rule is that minors can give independent consent from the age of 16. The only exceptions to this rule are when minors in the age group 16–18 years are to participate in clinical drug trials, or when bodily interventions are to be performed. Parental consent is only required in connection with these two exceptions. Therefore, we directly obtained informed consents from all participants, including those in the age group 16–18 years. In accordance with the ethical clearance for this study, we state to have obtained informed consent from a potential participant only when this participant has professed to understand the above-listed information and clearly stated her consent to participate. In our application for the ethical clearance, we have asked to allow oral consent alongside the written one to include illiterate participants and/or those having difficulties in giving written consent. The ethical committee approved our application without

stipulation for any form of documentation for the oral consent. Still, we voice-recorded the oral consents of all participants who agreed to that. None of the participants were paid for participation, but many were compensated for transport and/or wages lost in the form of a gift card (the equivalent of 30 USD).

The study was first approved in 2016 by the Norwegian Social Science Data Services as a pilot project, and then by the Norwegian Regional Committee for Medical and Health Research Ethics in 2017 as a Ph.D. project.

Recruitment was stopped once we ceased to observe new information in the collected data. 61 interviews were conducted with the 26 participants, as 17 participants were interviewed twice and nine participants three times. All interviews were conducted by the first author (MMZ) in participant-selected locations, such as cafés and homes of either MMZ or the participants themselves. The interviews with Sudanese participants were conducted in Arabic, while Somali participants were interviewed in either English or Arabic or a mixture of English and Norwegian.

The first interviews served two purposes: building trust, and inviting and encouraging the participants' own narrations of FGC related experiences. These interviews lasted between 30–90 minutes, during which neither voice-recording nor hand notes were taken. However, immediately after each interview, MMZ recorded her recollections, as well as her interpretations of the interview. In these interviews, MMZ asked the participants to narrate what they could remember and wanted to share concerning their circumcision experiences (See S1 Appendix. Semi-structured interview guide). They were also asked to focus during these narrations, wherever possible or applicable, on experiences of health problems they attributed to the circumcision, and how these health problems were dealt with and why. The participants' narrations went mostly uninterrupted and with a little probing. At the end of these first interviews, all participants were invited to second interviews. All of the 26 participants accepted the invitations and the time and location of the second interviews were agreed upon.

The second interviews were also flexible and lasted between 60–180 minutes. In these interviews, MMZ's interpretations and recollections of information given in the first interviews were individually discussed with the participants, probing for confirmations, corrections, and elaborations. Still, the need for probing was often minimal since the participants were mostly forthcoming with their views and experiences. All of the second interviews were voice-recorded with participants' permission.

Thereafter, the author's preliminary interpretations of patterns and emerging themes were individually discussed and expanded upon during additional third interviews with nine participants. These interviews were also voice-recorded and lasted between 60–90 minutes.

In addition, observations from MMZ's participation in over 20 workshops and seminars on FGC, as well as validation group discussions carried out at the end of data analysis with additional 17 Sudanese and Somali participants, added further insight and weight to the findings.

## Analysis

All voice-recordings, including those of MMZ's recollections and reflections after the interviews, were entered into NVivo 12. Recurrent themes and patterns were identified following the thematic analysis approach described by Braun and Clarke [51]. Still, rather than transcribing the voice-recordings, MMZ listened twice to each interview in its entirety before starting to assign codes to the different extracts of the audio. The initial codes were then refined in the light of new insights gained from the third interviews, as well as discussions with the last author (REBJ). Codes were then sorted into potential themes and all corresponding data extracts were collated under these initial themes. These themes were repeatedly reviewed and

refined until clearly identified themes and sub-themes were developed. The collated audio extracts under these refined themes and sub-themes were transcribed and translated directly to English by MMZ. Finally, using the theories of planned behavior and sexual script as additional lenses, the content of themes and sub-themes relevant to the research questions addressed in this article were consolidated into a rich and coherent description. To protect the participants' identity, and to minimize the risk of recognition, we have intentionally withheld and/or kept the participants' characteristics vague and ambiguous. We have also assigned more than one pseudonym to a few participants.

## Reflexivity

To build trust, the information given to the participants during recruitment was repeated in the first interviews, reemphasizing issues of confidentiality, de-identification, and possible repercussions for the researcher if she were to deviate from these ethical principles. Furthermore, we think MMZ's position and personality had helped to achieve a good balance between MMZ's roles as insider and outsider, as described by Kusow (2003), and had thus played a major role in gaining the trust of both the Sudanese and Somali participants. As a Sudanese female born and raised in Sudan, both Somali and Sudanese participants probably perceived MMZ as an insider i.e. someone who shared experiences and understanding of cultural subtleties and knowledge of FGC. Still, sharing the same background as the researcher could have the disadvantage of making some of the Sudanese participants afraid of gossip and judgment were they to share opinions or experiences that deviate from the community's socio-cultural norm. Fortunately, this did not seem to be the case. A likely explanation could be that MMZ's own deviations from the local Sudanese norms, demonstrated by her western style of dressing and her marriage to a Westerner, had led the participants to also regard her partly as an outsider. Furthermore, MMZ's medical background seemed to have further facilitated trust and openness. Some women said that MMZ's medical and cultural backgrounds made it easy to share information that they had not previously shared with anyone.

## Findings

Almost all participants explained their intention to use or not to use FGC-related healthcare services through references to different sets of social norms and expectations pertaining to pre-marital and marital sexual conduct. These social norms and expectations were often described by the participants as well-known, but mostly non-verbalized, sets of values that they were expected to conform to. All study participants described how they had learned, from early on, that marriage and motherhood were the measuring stick against which their future value would be assessed. They explained how an unmarried, divorced, and/or childless woman with an impressive repertoire of academic and/or professional achievements would commonly be referred to as "unfortunate", "unlucky", or even "a failure" by her family, friends and the local community. Therefore, women would generally be motivated to adhere to the norms that would increase their chances of getting and/or staying married. Nevertheless, it soon became evident that there were considerable variations among the participants regarding these latter norms, mainly those pertaining to staying married. These variations were evident between Somali and Sudanese, old and young participants, and within these groups. For example, a few Sudanese participants strongly believed that not displaying interest or pleasure in the sexual act was the sexual norm for married Sudanese women. Yet, this strong belief was intensely contradicted by other Sudanese participants. Hence, we found the term *subjective norms* as described in the theory of planned behavior [49] to best describe the varying perceptions among our participants on what they believed to be the societal sexual norms. Therefore, we

grouped similar subjective norms together, which eventually resulted in the formation of four different *cultural scenarios*. These scenarios included one premarital scenario (the virgin scenario) and three marital ones (the passive sexual partner scenario, the conditioned active sexual partner scenario, and the equal sexual partner scenario). Each of these cultural scenarios had implications for the acceptability of the various FGC-related healthcare options (See Table 2), mainly deinfibulation, clitoral reconstruction, and psychosexual counseling.

Although one could expect it to be easy to distinguish between the subjective norms forming the various cultural scenarios and the participants' own attitudes, we found that such distinction was only evident when there was a conflict or discrepancy between one's attitude and the subjective norm. Similarly, agency or perceived control was most evident in the narratives of those who rejected the subjective norms.

## The virgin scenario

All of the participants referred to virginity as the most important prerequisite for marriageability, both in their countries of origin and in their local communities in Norway. Also, almost all of the participants talked about sex or sexual intercourse in terms of penetrative vaginal intercourse, and of virginity in terms of the absence of such penetration. Virginity was claimed to be verifiable by the presence of an intact hymen. As long as girls were to avoid penetrative vaginal intercourse and consequently preserved their hymens, almost all of the participants believed that these girls would be considered virgins. On the other hand, girls were said to be considered as "open" and "non-virgins", even in the absence of any sexual activity, if their hymens were deemed not intact. It was also a widespread strong belief among the participants that during the first vaginal intercourse, a man could tell whether the girl's hymen was intact or not. A man would be expected to notice a number of physical signs that would either confirm or negate its intactness. These signs were described in relation to different stages of vaginal intercourse. The first sign would be a very tight vaginal opening sufficient to make the girl or woman scream out of pain upon the first attempts of penetration. The second was claimed to be a noticeable resistance from the hymen, which would temporarily prevent full vaginal penetration. The third and last sign was described as a snapping sound that would indicate the break of the hymen. Once the act was over, the ultimate proof would be the presence of blood, whether on the bedsheet, in-betweens the girl's or woman's thighs, or as droplets on the floor on her way to the bathroom. Many of the participants explained how these signs would be repeatedly emphasized in stories shared among friends about wedding nights. A few participants said that even though they had never engaged in premarital sexual activity, they were still very scared of failing to prove their virginity. They were aware of an ongoing discussion on the presence of different types of the hymen and that not all women would bleed at the first vaginal intercourse. One of them was Suha, a Sudanese participant in her early-20s who said:

> *"My married friends scared me. They told me that I will experience severe pain, will bleed and hear a pop sound . . . they really scared the hell out of me. . . In general, whether you bleed or not is a source of stress for both men and women. For the man, he will be very suspicious if the wife did not bleed. For the woman, if she did not bleed, she will be afraid that her husband will think that she was not a virgin. She will be really scared."*

Nevertheless, these physical signs (tightness, pain, and bleeding) were claimed not to always be sufficient. Almost all participants claimed that also women's behavior during first marital intercourse would be scrutinized for signs of virginity. Typically, a virgin should be shy, sexually ignorant, and very reluctant. Any deviation from these expectations was deemed sufficient

**Table 2. Summary description of themes and their implications for the intention to use FGC-related healthcare services and the care and prevention work on FGC.**

| Theme | Summary description* | Implications | |
|---|---|---|---|
| | | Intention to use FGC-related healthcare services | Care and prevention work on FGC |
| The virgin scenario | • Both Sudanese and Somali participants. <br>• As a sexual partner, a virgin: <br>○ Is shy, reluctant, and sexually ignorant. <br>○ Experiences severe pain upon her husband's first attempts of penetration, <br>○ Demonstrates that she is in pain (e.g. screaming), <br>○ Bleeds profusely once penetration is achieved (full vaginal penetration varies from few days up to several months). | Reluctance to undergo premarital deinfibulation. | • Concerns over safeguarding and proving virginity are central to the continued practice of FGC and the reluctance to seek healthcare. Activists and healthcare professionals should aptly address these concerns. |
| The passive sexual partner scenario | • Mainly Somali participants. <br>• Sexual intercourse is the husband's "legal right". Hence, it is the woman's duty to have sexual intercourse with her husband whenever he wants, as long as it is not anal sex, neither during her period. <br>• In addition, a good wife should: <br>○ Not show interest in sex (must not initiate), <br>○ Be passive (lie down like a log) and never show that she is enjoying sex even if that is the case. | Reluctance to use FGC-related healthcare services that focus on improving the women's sexual experiences such as psychosexual counselling and clitoral reconstruction. In contrast, deinfibulation in the context of pregnancy and childbirth is very acceptable. | • While assessing the sexual function of girls and women with FGC, health professionals need to remember that: <br>○ High frequency of sexual intercourse is not necessarily a reflection of the women's desire. <br>○ Not initiating sexual encounters is not necessarily a reflection of lack of desire. |
| The conditioned active sexual partner scenario | • Only Sudanese participants. <br>• Sexual intercourse is the husband's "legal right". Hence, it is the woman's duty to have sexual intercourse with her husband whenever he wants, as long as it is not anal sex, neither during her period. <br>• However, a good wife in this scenario is an active sexual partner. She: <br>○ Initiates (traditional or other signals: smoke bath, traditional perfume, sandal incense, sexy lingerie, and candle light). <br>○ Reciprocates and enjoys sexual intercourse with her husband–up to a limit (she should not exceed her husband's sexual stamina). <br>○ Demonstrates sexual desire and pleasure even when that is not the case. | FGC-related healthcare services that can improve the married women's sexual experiences such as psychosexual counselling and clitoral reconstruction are commonly accepted. | • While assessing the sexual function of girls and women with FGC, health professionals need to remember that: <br>○ High frequency of sexual intercourse is not necessarily a reflection of the women's desire. <br>○ Initiation is not necessarily a reflection of sexual desire. |
| The equal sexual partner scenario | • Mainly younger Somali and Sudanese participants in the age group 16–25 years. <br>• Virginity is irrelevant. <br>• Women's sexual pleasure have equal importance to that of men: <br>○ A girl or a woman can initiate, reciprocate and fully enjoy sexual intercourse. <br>○ She can also refuse her husband's/partner's sexual advances. <br>• Strongly believe that FGC is associated with sexual health complications. These beliefs were attributed to either first-hand knowledge from the girls' first experiences of sexual intercourse or to the media. This resulted in: <br>○ Feelings of shame and inadequacy. <br>○ Need to dissociate from the media image of circumcised girls as weak and sexually mutilated victims. | All FGC-related healthcare services that can improve the girls and women's psychosexual wellbeing such as deinfibulation, psychosexual counselling and clitoral reconstruction are strongly accepted. | • A need for critical reflections by activists and the media over the unintended repercussions of their choices of words and the contents of the messages they use in anti-FGC campaigns. |

*This summary description does not reflect the variations within each theme.

to raise the man's suspicion regarding the bride's virginity. Asha, a Somali participant in her mid-20s, explained why she pretended to be reluctant and sexually ignorant on her wedding night as follow:

> *"A friend of mine, her husband accused her of being sexually experienced. He told her he can see that she was excited. And that was just because . . . since she was educated and well-informed, she knew what was going to happen! So people tell you to pretend to know nothing, even if you do. You have to act stupid and naive and make him think that he is doing all the teaching."*

Many participants shared numerous stories about a few girls that they personally knew, and many that they did not, who were divorced immediately after the wedding nights because the husbands had declared them "used", "open" and "non-virgin". Such an outcome, to be declared as a non-virgin on the wedding night, was considered to be amongst the worst mishappenings that could transpire a Sudanese or a Somali girl and her family. It was described as "scandalous" and "bound to tarnish the family reputation once and for all". Most of the participants were hence of the opinion that Sudanese and Somali girls and women would feel tremendous responsibility towards preserving family honor.

The virgin scenario had a strong influence on girls' and women's perceptions of the acceptability of premarital deinfibulation. Regardless of the magnitude of health problems that unmarried girls and women might encounter because of infibulation, many of the participants claimed that they would be very skeptical of undergoing premarital deinfibulation. Many participants explained that the term *opening surgery*, the English translation of deinfibulation in Norwegian, would equate the procedure to "deflowering" and "compromising virginity". Since many participants considered it probable that future husbands would expect them to be virgins, they had felt it necessary to avoid such procedure. Sara, a Sudanese participant in her early-20s, who despite spending considerable time during the interview to describe how infibulation had adversely affected her health and school attendance, explained why she had decided to forgo premarital deinfibulation:

> *"I would say to myself 'what if I were to have the surgery now and then met a man and got married?' I won't be able to discuss this with him beforehand, you know . . . so if he found that it is not as he expected down there, he might say to himself 'this girl had slept with other men for sure!' Even if I explained, he might not believe me . . . he might even tell my parents and others that I was not a virgin! It would be a scandal! Just imagine that [uneasy giggles] after all my parents had done for me! . . . no, believe me, I am better off this way."*

Furthermore, many participants considered preserving the integrity of the infibulation seal to be a sure measure to pass as virgin. To enable penetrative vaginal intercourse in an infibulated girl or a woman, the seal of skin, ensuing from the narrowing of the vaginal opening during infibulation, would first have to be torn or cut open. This would commonly be done by the husband, using his penis, finger, or other tools over a period of time varying from few days up to several months. This process would cause the girl or woman to bleed and scream from pain. Thus, it would seem that the typical response of an infibulated girl or woman to her first experiences of penetrative vaginal intercourse had become an overarching expectation of how virginity should manifest. Also, this overarching expectation would appear to have become the social script for virginity for all women, irrespective of FGC. Sure enough, many participants considered it important to avoid premarital deinfibulation so as to conform to the social expectations of a virgin. However, some of the participants had different attitudes towards virginity than the subjective norms. These participants were very critical towards the virginity

expectation and refused to be measured by it. Two of these participants argued that virginity was meaningless since a girl could engage in a wide variety of sexual activity, including anal intercourse, and still be considered a virgin. Other participants argued that since virginity in Islam was a requirement for both the husband and the wife, and since husbands could not physically prove their virginity, women should similarly not be asked to prove it. The rest described how they, early in life, had decided whether they intended to marry a fellow country-man, and how such a decision had subsequently influenced how they valued their virginity. This quote from Khadija, a Somali participant in her early-40s who had undergone premarital deinfibulation as a 19-years-old, would be a good example:

*"I was never concerned about virginity. Since I was a child, I kept telling my mum that I will never marry a Somalian man. It did not even have anything to do with virginity or circumcision at the time. I just did not have anything in common with them. My way of thinking and their way of thinking was totally different. I always said that. . . and because I said it many times it did not come as a shock when I decided to marry a Norwegian. But even if I were to marry a Somalian, I believe that there are modern Somalian men who do not care whether you are open or not."*

The perceived control over the decision of not to marry a fellow countryman, unless he was modern, would seem to have influenced Khadija and a few other participants to no longer care about preserving the proof of their virginity. Such perceived control would also appear to have made it simpler for them to undergo or to consider undergoing premarital deinfibulation.

## The passive sexual partner scenario

In the first of the three cultural scenarios pertaining to marital sexual conduct, a good wife was described as one who should be sexually available, yet passive. The subjective norms that informed this scenario were shared more by the Somali participants than the Sudanese. Many participants, adhering to both this scenario and the next one, stressed the husband's "legal right" to sexual intercourse. Hence, they considered it to be the woman's duty to let her husband "do his thing" according to his wishes and regardless of hers. However, there were a few exceptions. Many participants cited the husbands' wishes for anal intercourse or sexual intercourse during menstruation as legitimate reasons for the women to not obey the husbands, as both acts were claimed to be forbidden in Islam (*haram*). A few also considered severe illnesses a legitimate reason, but other participants strongly disagreed. Except for these reasons, these participants alleged that it was a wife's duty to patiently await and never reject her husband's advances. Many of the participants believed that if a wife were to deny her husband, she would be cursed by the angels. The notion of marital rape was absent from the descriptions of all but one of the participants. This participant shared the story of a Sudanese friend who claimed to be raped by her husband during their honeymoon, even though the marriage was the result of a two-years-long love story. After tolerating the husband's initial attempts at deinfibulation for several days, the excruciating pain suffered during these attempts had eventually led her to refuse further attempts. The husband had thus become impatient and tied her up and forced himself on her. This led the friend to ask for a divorce and to disclose the reason for this decision to close family members and friends. However, instead of being met with understanding, she was told that this was not rape, but rather a justifiable means for the husband to exercise his legal right. Subsequently, she was pressured to return to the husband with strong instructions to never deny him again. This story would seem to indicate that sexual gratification was largely considered a man's prerogative. An indication that was further strengthened by the

acceptance of many participants that women should refrain from initiating sexual intercourse as this would imply that women have sexual desire. The latter was believed to scare the husbands as they would think that their wives had "excessive" sexual urges that might make them more inclined to cheat. Hawa, a Somali participant in her mid-20s, described how the husband of a friend had tried to solve what he allegedly considered problematic behavior:

*"Men do not like women to take the initiative. They ask the women what is wrong with them. I know someone who was told by her husband not to eat chocolate ever again . . . he thought that chocolate was what made her ask for sex!"*

Many participants explained that out of fear of raising the husband's mistrust, they would refrain from initiating sexual intercourse. For the same reason, if the women who adhered to this sexual scenario were to feel sexual enjoyment during the sexual act, they would make sure to hide it. They generally portrayed sexual intercourse as something not to be enjoyed by the women but rather endured. Still, almost all of these participants said that the prospect of having children made sexual intercourse worth the trouble. The following quote from Fathia, a Somali participant in her mid-30s, was typical:

*"Sex is not something to enjoy. It is to have babies. The man will come, do his thing and then leave you alone. He does not care that you did not enjoy it. Some men will not trust you if they think you like sex."*

Overall, almost all of those who identified with this scenario said that healthcare services aiming to improve sexual health, specifically sexual counseling and clitoral reconstruction, were totally unsuitable for them. A typical response to inquiries regarding why they have not sought help for their reported experiences of sexual pain or lack of sexual gratification, was giggles that were eventually replaced by serious facial expressions and comments similar to the following by Halima, a Somali participants in her late-20s:

*"Wouldn't it have been great to finally experience it [orgasm]? [Giggles] But seriously, no, I am happy that it [sexual intercourse] finishes quickly. I am usually so tired and just want to sleep . . .you know . . . after shopping for groceries, cooking, cleaning, doing laundry, looking after the children . . . I am the one who has to do everything around here, you know . . . I just want to be left alone . . . honestly, I would be happy if he found someone else to give him sex [Giggles] No way I would consider changing anything . . . even now [when I am] not enjoying sex, my husband is often suspicious! Are you crazy? Of course, I wouldn't!"*

Thus, healthcare services to improve sexual pleasure for the women such as clitoral reconstruction and psychosexual counseling were considered inherently incompatible with the requirements of their roles as wives. Also, many of the participants described relationships characterized by emotional detachment and painful sexual encounters that made them shy away from intimacy and/or sexual intercourse. Marital deinfibulation, on the other hand, was viewed more positively by these participants but only in the context of pregnancy and childbirth. In other words, unlike improved sexual life, childbirth was considered as an acceptable reason to undergo deinfibulation.

### The conditioned active sexual partner scenario

Akin to the previous cultural scenario, this second marital scenario, described only by Sudanese participants, considered sexual intercourse and sexual gratification as the husband's legal

right and prerogative. However, expectations to the wives' sexual behaviors differed in that they had to be actively engaged. Almost all participants adhering to this scenario agreed that once virginity was proven, a good wife should initiate, reciprocate, and display enjoyment during the sexual intercourse. Yet, there seemed to be a ceiling to this active engagement, which was mostly explained in terms of not exceeding or overtaxing the husband's sexual drive.

Overall, sexual intercourse was talked about as a cornerstone for marriage that if neglected, marriage would crumble. Wives would be warned that if they did not cater to their husbands' sexual needs, their husbands would most probably cheat, take a second wife and/or divorce them. This perception was expressed in many ways. For example, a woman complaining of an unfaithful husband would be told that it was her fault as she did not please him sexually. Also, if a wife complained about her husband's negligence, stinginess, or aloofness, she would be told that sexual enticement would be her trump card and the way to go. This sexual enticement was described by the participants, adhering to this scenario, in terms of a variety of in- and out- of bed methods. Out-of-bed methods included: wearing sexy lingerie, creating a romantic environment at home, wearing a henna tattoo, waxing, and using traditional sauna (*dukhan*), incense (*bakhoor*), perfumes (*khumra*), and scrubs (*dilka*). In-bed methods included: being attentive and accommodative to the husband's sexual needs and fantasies, as well as displaying sexual pleasure through body movements and sounds of pleasure. Most of these participants professed to find the out-of-bed methods easy to adopt. In contrast, several had problems with methods pertaining to the in-bed enticement. These latter participants mostly attributed the sexual problems they were facing, such as lack of desire and sexual excitement, pain, and inability to reach orgasm, to circumcision. Some claimed that their ability to enjoy sexual intercourse was cut away together with their clitorises. A few also said that the narrow vaginal opening, ensuing from infibulation, meant that sexual intercourse was a continuous source of pain even after years of marriage. Thus, to still be expected to experience and display sexual pleasure seemed for many of them unfair and a clear double standard. They further explained that the reason they had been subjected to FGC, in the first place, was the perceived need of meeting the Sudanese men's requirement of virginity. Subsequently, they blamed the men for all sexual challenges they were experiencing and felt "betrayed" and "disappointed" when their spouses' accused them of being "frigid" and "cold". To avoid being called such names, as well as to sexually please their husbands, several participants claimed that brides would be advised by other married friends or relatives to use lubricants and local analgesics, hide their difficulties, and fake desire and orgasm. The following quote from Nagla, a Sudanese participant in her early-20s, depicted some of the aforementioned expectations of a good wife according to this cultural scenario:

> "The intimate relation is essentially an interaction where the two melt into each other. Sudanese men, unless they are cold, do not like cold women. So, as a wife, I should show my interest and make myself ready. As you know, once you have the dukhan, henna, khumra, or dilka, your man will know you are ready... Of course, some women do that out of duty and then lie down there like a log. This is so bad! Interaction and reciprocation make the marriage stronger. If I was cold and he met a beautiful woman in the street, he will be seduced by her. So, if I was cold, I will have to fake it to keep my home and marriage intact."

Almost all of the participants, adhering to this scenario, claimed that sexual interest and desire were easy to fake. Yet, faking orgasm was found by some participants unfamiliar and difficult to the extent that a few had actually received detailed instructions from close friends and relatives on how to do that. A typical example of such instructions was given by Nada, a Sudanese participant in her 40s:

*"She told me to move, increase the rhythm of my breathing and make it louder . . . also . . . to grunt and to contract my vagina . . . it was very graphic! I was shocked."*

Although almost all of the participants believed faking to be a good strategy, those who professed to do that were a bit skeptical. Some felt that it often made their spouses spend a long time experimenting with new sexual positions, which meant they had to endure for a longer time than if they were not pretending. Many also experienced faking sexual excitement as emotionally draining. They confessed to feelings that resembled grief and longing, grief over the cut clitoris, and longing for these sexual feelings that they were faking. The following quote from an interview with Munira, a Sudanese participant in her 30s, would illustrate how these participants typically talked about the clitoris and sexual pleasure:

*"I was thinking about it [the clitoris] all the time. I will be in the cinema and suddenly I am looking around wondering if I am the only one there without a clitoris. . . the only one who doesn't fully enjoy sex. So, I was very excited when I first heard of clitoral reconstruction. It was on a TV documentary I think. . . I am not sure. . . anyway, for months I couldn't think of anything else! I asked all my friends if they knew anything about it. I even called a doctor friend in Sudan. . . he is actually a man! Imagine how shameless I was [Laugh] it shows you how desperate I was then [Giggles]. He told me he didn't know about the operation. He wasn't a gynecologist, that's why. He thought it might be useful, but he thought what I really needed was counseling. . . he thought the real problem was in my head. . . he was probably right. . . anyway, I was very disappointed when I found out that it [clitoral reconstruction] wasn't available in Norway."*

The majority of our Sudanese participants seemed to be very interested in improving their sexual lives and, thus, in healthcare help in the form of psychosexual counseling and clitoral reconstruction. This finding was also emphasized by all of our six key informants, who worked closely with a much larger number of Somali and Sudanese women than the one we have interviewed. The following quote by Sara, one of these key informants, was typical:

*"You know, I worked for a long time with the Somali [before also started working with the Sudanese] and was used to the fact that they won't discuss sex in front of others. So I was in a way unprepared for the Sudanese! [Giggles] The way they openly talked about sex. . . it was refreshing! [. . .] They told me many times that they needed to talk to a professional who can teach them about sex and sexual pleasure."*

Still, the participants' apparent interest in improving their sexual lives had not translated into an unequivocal interest in marital deinfibulation. Instead, the participants' attitudes towards marital deinfibulation were primarily influenced by their husbands' preferences rather than the subjective societal norms. A few considered, in agreement with their spouses, their experiences with marital deinfibulation to be very positive. These deinfibulations had commonly been undertaken to facilitate vaginal penetration and/or childbirth. They said to feel a noticeable improvement in their sexual lives, both in terms of reduced pain and increased sexual pleasure after deinfibulation. In contrast, other participants had negative attitudes towards marital deinfibulation. These participants were unhappy with the outcome of their deinfibulation experiences, which had been performed in connection to pregnancy and childbirth. These participants insisted that the doctors had opened them more than what they believed to be necessary and appropriate. A major concern was their spouses' negative reaction. They complained about how their spouses were repeatedly telling them that they had become "too wide",

using idiomatic expressions to equate vaginal penetration after deinfibulation to driving in "the highway". Subsequently, these participants were seriously contemplating options that would make them "tight" once again such as using the special herbal mix "*Afsa*" or even traveling abroad to undergo reinfibulation. On the other hand, three participants had gone against their husbands' wishes for marital deinfibulation. These participants revealed that they had strongly resisted their husbands' pleas to seek deinfibulation during their honeymoons. Fear of reliving their childhood experiences of infibulation, motivated them to take control and refuse to conform to their spouses' wishes, even at the high expense of getting divorced.

## The equal sexual partner scenario

In this last scenario, women's sexual pleasure was believed to have equal importance to that of men. It was mainly younger Somali and Sudanese participants in the age group 16–25 years who helped to inform this scenario, but a few were much older. These participants seemed to identify with subjective norms that centered on the ideals of gender equality. They expected their boyfriends or husbands to contribute equally to household chores and the raising of children, and in return that they (the participants) would contribute financially. Also, that they would be free to initiate, reciprocate and fully enjoy and express their sexuality. The following quote from Aminah, a 21 years old Somali participant was typical:

> "*We are in Norway*! *This is not Somalia, where the woman has to do everything . . . in and out of the house. Even in the most conservative countries . . . like Saudi Arabia . . . it is expected that men share some of the responsibilities. They take care of money and heavy work, while the women stay at home to clean, cook, and take care of the kids. But in Somalia, the women will do everything and the men nothing*! *Luckily we are in Norway now. We do everything equally [smile].*"

Almost all of the participants identifying with the subjective norms on gender and sexual equality attributed these norms to the prevailing values in the larger Norwegian society. The exception was a couple of participants who attributed these values to what they claimed were the correct Islamic teachings. They clarified that according to the correct Islamic teachings, sexual gratification was an equal legal right of the wife as it was of the husband. Therefore, it was also a husband's duty to sexually please his wife.

Furthermore, most of the participants identifying with this scenario anticipated full autonomy in all personal decisions, including the right to reject unwanted sexual advances, even those made by boyfriends and husbands. Thus, these participants strongly considered it rape if they were to be subjected to forced sexual advances by a boyfriend or a husband, and found it unforeseeable that family members or friends would try to convince them otherwise. A few of these participants also seemed to have strained relationships with their families. They confessed to harbor strong feelings of anger towards their parents, particularly their mothers, for failing to protect them from FGC. They believed FGC to be a "barbaric" and "cruel" tradition that negatively affected their health and wellbeing. Still, they chose to suppress this anger instead of talking it over with either their parents or professionals. They found talking to their parents about such feelings to be extremely difficult. Similarly, they were afraid that talking to professionals would exacerbate prejudices towards parents of non-Western backgrounds. The rest of participants had equally negative attitudes towards FGC, but they claimed to have forgiven their parents for subjecting/ letting them be subjected to FGC. They generally excused their mothers for "not knowing better" or "being really young" at the time. Also, common to the participants represented by this scenario was their strong beliefs concerning sexual health

complications of FGC and their positive attitudes towards healthcare services geared towards improved sexual health. Five of the participants explained that their strong beliefs on sexual complications stemmed from first-hand knowledge gained during their first experiences of sexual intercourse. These five participants clarified that they had found it difficult to disclose, beforehand, their FGC status to their boyfriends/husbands. Therefore, they ended up in awkward situations when, amid their first sexual encounters, it was suddenly revealed that they were infibulated. Two of them had dealt pragmatically with the situation, while the other three professed to still struggle with residual feelings of shame. One of the two participants who took a pragmatic approach was Jawaahir, a Somali participant, who was 19 years old at the time:

> *"We tried to have sex. He tried and tried but could not get it in . . . On top of that, it was extremely painful. So I went to my general practitioner and told him what has happened. He referred me to the women clinic to get the surgery [deinfibulation]. We tried again sometime after the surgery and it worked perfectly this time. Mission accomplished, case closed!"*

For Jawaahir and another participant, the awkward and difficult first sexual encounters did not result in residual feelings of inadequacy or shame. Both claimed that deinfibulation was all they needed to move forward with their sexual and social lives. In contrast, Ilham, a 27 years old Sudanese participant, was one of those who professed to continue to struggle with residual feelings of shame and poor self-image many years afterward:

> *"It was like a nightmare! The wedding party was over and we went to the hotel. And there we were . . . the two of us alone. Like any other bride, I was shy, yet excited. It started well . . . but suddenly in the middle of it all, he stopped . . . he got up . . . and got dressed . . . I was confused. He sat next to me and covered me with the bedsheet. He then said to me 'you are infibulated, right? Why didn't you tell me? I never wanted to marry a girl that was circumcised, let alone infibulated! Still, if I knew you were infibulated, we could have arranged for it to be opened'! I was mortified. I wished the ground would open and swallow me up! I told him that it wasn't too late; he can still divorce me and find another one who is not circumcised to marry. He hugged me and reassured me that was not what he meant . . . he said he loved me, and because he loved me he didn't want to cause me any pain . . . and that would have been impossible with infibulation. So if I had told him beforehand, he could have arranged for it [infibulation] to be opened. I eventually calmed down and agreed to go and have it opened by a doctor. But you know what? Something inside me was broken that night! I feel so disgusted with myself! He tries to reassure me and tell me that now I am normal down there, but I won't let him look. I always turn the light off when we make love".*

For Ilham and the other two participants, deinfibulation was not sufficient on its own. All three believed that they would benefit from psychosexual counseling. One of them also contemplated traveling to Sweden to reconstruct her clitoris to "completely reverse the circumcision".

The remaining participants who adhered to this scenario said to have first acquired their knowledge of FGC-related sexual complications from the media. All of these participants expressed deep feelings of shame and a pressing need to dissociate from the image presented in the media of circumcised girls as weak and sexually mutilated victims. One of these participants kept her FGC status a top-secret and decided to avoid any situation that could expose her as "mutilated". She claimed that she was unaware of any FGC-related healthcare services. Hence, she had reluctantly accepted the idea of a lonely and childless future. The other participants also kept their FGC status as a secret but decided to act proactively to minimize the risk of being exposed as "mutilated". These participants underwent deinfibulation a long time before

they were sexually active. Two of them found that they had their clitorises intact underneath the infibulation, which made it very easy to pretend that they had never been circumcised. Nevertheless, the other two participants, of whom one was married to a Norwegian, said that even after deinfibulation they were still unable to hide their FGC status. Therefore, when they started to have problems reaching orgasm, they decided to fake it rather than risk drawing attention to their original FGC status. Yet, they were unhappy with the toll faking orgasm had taken on both their health and relationships. They professed to find it frustrating that psychosexual counseling was not offered alongside deinfibulation in the specialized FGC-healthcare clinics.

## Discussion

With the help of the theory of planned behavior as well as the sexual script theory as additional analytical lenses, we explored the intention of Somali and Sudanese immigrants in Norway to use a number of FGC-related healthcare services.

We found that almost all of the subjective norms that our participants identified with, and which pertained to the intention to use FGC-related services centered on getting and/or staying married. These subjective norms represented different outcomes of the interplay between norms in countries of origin, norms in Norway and the expectations and attitudes of the husbands and partners. The latter was typically gleaned through interpersonal interactions. The different outcomes of this interplay seemed to have changed gradually among the Sudanese participants from traditional subjective norms, to in-between or transitional ones, and finally to subjective norms of gender equality. Contrary to earlier findings from Sudan [52, 53], only small numbers of our Sudanese participants exclusively identified with the traditional norms of virginity and passive sexual behavior. The majority of our Sudanese participants related to transitional subjective norms that combined traditional norms of chastity and virginity with norms of active sexual behavior. A few rejected all traditional subjective norms and fully adopted those of sexual and gender equality. While no other study did describe these transitional subjective norms, other studies in Norway [9, 48], Canada and Australia [54] similarly identified virginity as an important sexual norm for their Sudanese participants. For our Somali participants, the outcomes of the interplay between the different norms varied between total support of traditional subjective norms such as virginity and passive sexual behavior among older participants to complete rejection of the same norms among many of the younger participants. These findings were in accordance with other findings from Norway [9, 48], the United States [55], Canada and Australia [54]. While the concept of sexual intercourse as a duty and obligation was linked to traditional norms in our participants' countries of origins, this concept, together with other sexual challenges such as faking orgasm, were also documented among ethnic Norwegians [56] and American women [57] who did not practice FGC, but suffered from decreased sexual desire, sexual pain, and anorgasmia.

Our second finding was that each of the traditional and transitional subjective norms, as well as the subjective norms of gender equality, had specific implications for the intention to use FGC-related healthcare services. Similar to previous findings [9, 48], we found traditional subjective norms, which we grouped under the virgin and passive sexual partner scenarios, to have negative implications for the intention to use almost all of the FGC-related healthcare services. In contrast, subjective norms of gender equality had positive implications for the intention to use all of the same services. The picture was more complex though for the transitional subjective norms that we grouped in the conditioned active sexual partner scenario. We found the transitional subjective norms to have mostly positive implications for the intention to use services that help to improve marital sexual lives, yet negative implications for the intention to use premarital services.

We further found that attitudes and subjective norms pertaining to the intention to use FGC-related services were unsurprisingly inseparable when the participants found these norms agreeable or unavoidable. However, it was possible to distinguish between the two when there was a discrepancy between the participants' attitudes and the subjective norms. Finally, we established that agency/control was exercised to both reject and conform to negative subjective norms pertaining to the intention to use FGC-related services. Some participants had undergone premarital deinfibulation and risked dishonoring their families by rejecting the negative subjective norms towards the procedure, while others refrained from undergoing marital deinfibulation and risked divorce by rejecting the positive subjective norms towards this later procedure. Hence, reluctance to utilize FGC-related healthcare services should not be equated with a lack of agency.

## Conclusion

In this article, we explored the intention of Somali and Sudanese immigrants in Norway to use a number of FGC-related healthcare services. We found that the intention to use these healthcare services varies between and within our two groups of Somali and Sudanese participants. We also found that the intention to use different services, or even the same service but at different points of time, could vary in the same individual. Nonetheless, many of our participants have positive attitudes towards psychosexual counseling and would most probably use the service were it to be offered as part of the Norwegian FGC-specialized healthcare services. These findings indicate that the underutilization of some FGC-related healthcare services does not necessarily mean that other FGC-related services would also be underutilized. An insight that could prove valuable not only for Norwegian but also for other policy-makers and healthcare professionals during the planning and/or delivery of FGC-related healthcare services. In addition, our findings give deeper insights into the meaning of FGC for immigrant women. One of these insights is that infibulation acts as a safeguard and/or evidence of virginity for girls with FGC, hence many of these girls are reluctant to seek and/or accept healthcare. These concerns over safeguarding and/or providing proof of virginity could also contribute to the continued practice of FGC. Therefore, both activists and healthcare professionals should aptly address these concerns. Furthermore, as evident in this article, sexuality and FGC are closely related. Hence, in their meeting with girls and women with FGC, healthcare professionals should not avoid talking about sexuality. They should also remember that many girls and women with FGC perceive sexual intercourse with their husbands as a religious duty that they cannot refuse. Therefore, while assessing the sexual function of these girls and women, healthcare professionals should not equate the high frequency of sexual intercourse with these women's desire. Similarly, initiation or lack of initiation of sexual intercourse by these girls and women is commonly governed by sexual norms and should not be equated with either desire or lack of desire. These latter insights could also be valuable for the validation of instruments used to assess the female sexual function. Finally, our findings indicate an urgent need for critical reflections by activists, the media and healthcare professionals over the use of terminology and messages that can contribute to feelings of shame and inadequacy in young girls. Many young participants in the age group 16–25 years harbored feelings of shame and inadequacy and a need to dissociate from the prevailing image of circumcised girls as weak and sexually mutilated victims.

## Supporting information

**S1 Appendix. Semi-structured interview guide.**
(PDF)

## Acknowledgments

We would like to thank our key informants for their insightful reflections and valuable help with the recruitment of participants.

## Author Contributions

**Conceptualization:** Mai Mahgoub Ziyada, Inger-Lise Lien, R. Elise B. Johansen.

**Data curation:** Mai Mahgoub Ziyada.

**Formal analysis:** Mai Mahgoub Ziyada.

**Funding acquisition:** Mai Mahgoub Ziyada, Inger-Lise Lien, R. Elise B. Johansen.

**Investigation:** Mai Mahgoub Ziyada.

**Methodology:** Mai Mahgoub Ziyada.

**Project administration:** Inger-Lise Lien.

**Supervision:** R. Elise B. Johansen.

**Validation:** Mai Mahgoub Ziyada, R. Elise B. Johansen.

**Writing – original draft:** Mai Mahgoub Ziyada.

**Writing – review & editing:** Mai Mahgoub Ziyada, Inger-Lise Lien, R. Elise B. Johansen.

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
