## [Decision Letter · Decision Letter 0]

8 Jan 2020

PONE-D-19-30005

Cultural scenarios pertaining to premarital and marital sexual conduct and the intention to use healthcare services related to Female Genital Cutting: a qualitative study among Somali and Sudanese women in Norway

PLOS ONE

Dear Dr. ziyada,

Thank you for submitting your manuscript to PLOS ONE. After careful consideration, we feel that it has merit but does not fully meet PLOS ONE’s publication criteria as it currently stands. Therefore, we invite you to submit a revised version of the manuscript that addresses the points raised during the review process.

We would appreciate receiving your revised manuscript by Feb 22 2020 11:59PM. To enhance the reproducibility of your results, we recommend that if applicable you deposit your laboratory protocols in protocols.io, where a protocol can be assigned its own identifier (DOI) such that it can be cited independently in the future. For instructions see: http://journals.plos.org/plosone/s/submission-guidelines#loc-laboratory-protocols

We look forward to receiving your revised manuscript.

Kind regards,

Nülüfer Erbil, Ph.D, Prof.

Academic Editor

PLOS ONE

Journal Requirements:

2. Please amend your current ethics statement to address the following concerns: Please explain why was written consent was not obtained, how you recorded/documented participant consent, and if the ethics committees/IRBs approved this consent procedure.

3. You indicated that you had ethical approval for your study. In your Methods section, please ensure you have also stated whether you obtained consent from parents or guardians of the minors included in the study or whether the research ethics committee or IRB specifically waived the need for their consent.

4. Please include a copy of the interview guide used in the study, in both the original language and English, as Supporting Information, or include a citation if it has been published previously.

Reviewers' comments:

Reviewer's Responses to Questions

**Comments to the Author**

1. Is the manuscript technically sound, and do the data support the conclusions?

Reviewer #1: Yes

Reviewer #2: Yes

2. Has the statistical analysis been performed appropriately and rigorously? 

Reviewer #1: N/A

Reviewer #2: No

3. Have the authors made all data underlying the findings in their manuscript fully available?

Reviewer #1: No

Reviewer #2: Yes

4. Is the manuscript presented in an intelligible fashion and written in standard English?

Reviewer #1: Yes

Reviewer #2: No

5. Review Comments to the Author

Reviewer #1: Thank you for the opportunity to review this paper. I really enjoyed reading it. The topic is of key importance for understanding migrant women’s views and experiences around FGC and related health service development. The paper is unusual in the richness of data and cultural understanding presented. I particularly liked the use of the theories (sexual scripts & theory of planned behaviour). The methodology and methods are well described and justified. I have no major comments on most sections of the paper except for the final ones (Discussion/Conclusion). Here, I feel it would be useful to highlight more explicitly what the implications are of the research for health service development and community engagement around FGC. For example, in the current Conclusion, there is a statement as follows: “This apparent unfulfilled need and other important insights provided in this article regarding subjective norms that could hinder/facilitate the utilization of FGC related healthcare services could prove valuable not only for Norwegian but also for other decision-makers and healthcare professionals during the planning and/or delivery of FGC-related healthcare services…..” I feel that the phrase “other important insights” warrants further elaboration. These are implicit in the Discussion but need to be made more explicit. I would suggest making your key insights clear (re-state them) and then link them to specific implications and recommendations. I also wonder whether you could comment on whether you think your research has deepened our understanding more generally of the meaning of FGC for migrant women – compared with the way it is represented in more traditional biomedically oriented health services research? It would be nice to have a commentary and critique of how this work may help to frame western thinking in this area in a different way. And what are the implications of this for future research – and for prevention related efforts?

The paper is generally very well written. Does the abstract need sub-headings?

Reviewer #2: The topic of this study is orginally. To increase intelligibility of the themes in article, themes may show as a table.

It sholud be write the validity and reliability of this study more detaily in material and method.

6. PLOS authors have the option to publish the peer review history of their article (what does this mean?). If published, this will include your full peer review and any attached files.

Reviewer #1: Yes: Dr Catrin Evans

Reviewer #2: No

---

## [Author Response · Author response to Decision Letter 0]

10 Feb 2020

We would like to thank the academic editor and the reviewers for their valuable time and feedback. 

In this letter, we provide point-by-point response to both the editor and reviewers.

Response to the academic editor

http://www.journals.plos.org/plosone/s/file?id=wjVg/PLOSOne_formatting_sample_main_body.pdf and http://www.journals.plos.org/plosone/s/file?id=ba62/PLOSOne_formatting_sample_title_authors_affiliations.pdf.

Response: done.

2. Please amend your current ethics statement to address the following concerns: Please explain why was written consent was not obtained, how you recorded/documented participant consent, and if the ethics committees/IRBs approved this consent procedure.

Response: done. Also, amended in the methods section, Lines 189-196.

3. You indicated that you had ethical approval for your study. In your Methods section, please ensure you have also stated whether you obtained consent from parents or guardians of the minors included in the study or whether the research ethics committee or IRB specifically waived the need for their consent.

Response: In this study, we did neither obtain consent from parents or guardians of the minors (16-18 years old) included in the study, nor asked the ethical committee for a waiver of parental consent. According to the Norwegian Health Research Act (https://lovdata.no/dokument/NL/lov/2008-06-20-44), the general rule is that minors can give independent consent from the age of 16. The only exceptions to this rule are when minors in the age group 16-18 years are to participate in clinical drug trials, or when bodily interventions are to be performed. Parental consent is only required in connection with these two exceptions. This clarification is now added to the methods section lines 183-189.

4. Please, include a copy of the interview guide used in the study, in both the original language and English, as Supporting Information, or include a citation if it has been published previously.

Response: done. The original language for our semi-structured interview guide is English. This interview guide is now included as “S1 Appendix. Semi-structured interview guide”.

Response to reviewer #1

1. Thank you for the opportunity to review this paper. I really enjoyed reading it. The topic is of key importance for understanding migrant women’s views and experiences around FGC and related health service development. The paper is unusual in the richness of data and cultural understanding presented. I particularly liked the use of the theories (sexual scripts & theory of planned behaviour). The methodology and methods are well described and justified. I have no major comments on most sections of the paper except for the final ones (Discussion/Conclusion). 

a. Here, I feel it would be useful to highlight more explicitly what the implications are of the research for health service development and community engagement around FGC. For example, in the current Conclusion, there is a statement as follows: “This apparent unfulfilled need and other important insights provided in this article regarding subjective norms that could hinder/facilitate the utilization of FGC related healthcare services could prove valuable not only for Norwegian but also for other decision-makers and healthcare professionals during the planning and/or delivery of FGC-related healthcare services…..” I feel that the phrase “other important insights” warrants further elaboration. These are implicit in the Discussion but need to be made more explicit. I would suggest making your key insights clear (re-state them) and then link them to specific implications and recommendations. 

Response: done. Lines 692-698, and 700-704.Also in table 2.

b. I also wonder whether you could comment on whether you think your research has deepened our understanding more generally of the meaning of FGC for migrant women – compared with the way it is represented in more traditional biomedically oriented health services research? 

Response: done. Lines 706-722.

c. It would be nice to have a commentary and critique of how this work may help to frame western thinking in this area in a different way. And what are the implications of this for future research – and for prevention related efforts?

Response: done. Lines 722-726.

2. The paper is generally very well written. Does the abstract need sub-headings?

Response: We have revised the abstract and used sub-headings. Lines 16-62.

Response to reviewer #2:

1. To increase intelligibility of the themes in the article, themes could be presented in a table

Response: done. A summary description of the themes is now included as table 2. Lines 292-294.

2. Should write/discuss validity and reliability in material and method.

Response: In this article, we followed the consolidated criteria for reporting qualitative research (COREQ) checklist. We have also systematically addressed issues related to the quality of qualitative data (credibility, transferability, dependability and confirmability), which relatively corresponds to internal and external validity, and reliability. For example, in the methods and/or findings sections:

• To enhance dependability (reliability) and transferability (external validity), we provided comprehensive record and thick description of:

The theoretical framework, Lines 128-156.

Sampling, recruitment, and study participants, Lines 169-198. 

Setting and data collection, Lines 202-231.

Data management and analysis, Lines 233-247. 

Themes and sub-themes that we systematically illustrated by participants’ quotations, Lines 300-393, 394-455, 456-554, and 555-641.

• To enhance credibility (internal validity), we:

a) Interviewed the same participant more than once, thus allowing for member checking:

 - In second interviews, the first author discussed with each participants her interpretations and recollections of information given in the first interviews, probing for confirmations, corrections, and elaborations. Lines 220-223.

 - In the third interviews, the first author expanded upon and discussed individually with nine participants her preliminary interpretations of patterns and emerging themes. Lines 226-227. 

b) Validated the findings in three group discussions with additional 17 Sudanese and Somali participants. Lines 229-231.

c) The first author repeatedly discussed codes and themes with the last author (REBJ). Hence, the ensuing interpretations and findings became a negotiated version of more than a one person’s frame of reference. Lines 237-238.

• To enhance confirmability, we provided detailed information on the personal characteristics of the interviewer (the first author) and her relationship with the participants under the sub-heading “reflexivity”. Lines 248-265.

Yours sincerely 

Mai Mahgoub Ziyada

MSc, MPhil and MBBS

PhD Candidate

Section for Trauma, catastrophes and forced migration - NKVTS

Institute of health and society, Faculty of Medicine - UiO

m.m.ziyada@nkvts.no

---

## [Decision Letter · Decision Letter 1]

8 Apr 2020

PONE-D-19-30005R1

Sexual norms and the intention to use healthcare services related to Female Genital Cutting: A qualitative study among Somali and Sudanese women in Norway.

PLOS ONE

Dear Dr. ziyada,

Thank you for submitting your manuscript to PLOS ONE. After careful consideration, we feel that it has merit but does not fully meet PLOS ONE’s publication criteria as it currently stands. Therefore, we invite you to submit a revised version of the manuscript that addresses the points raised during the review process.

We would appreciate receiving your revised manuscript by May 23 2020 11:59PM. To enhance the reproducibility of your results, we recommend that if applicable you deposit your laboratory protocols in protocols.io, where a protocol can be assigned its own identifier (DOI) such that it can be cited independently in the future. For instructions see: http://journals.plos.org/plosone/s/submission-guidelines#loc-laboratory-protocols

We look forward to receiving your revised manuscript.

Kind regards,

Nülüfer Erbil, Ph.D, Prof.

Academic Editor

PLOS ONE

Reviewers' comments:

Reviewer's Responses to Questions

**Comments to the Author**

1. If the authors have adequately addressed your comments raised in a previous round of review and you feel that this manuscript is now acceptable for publication, you may indicate that here to bypass the “Comments to the Author” section, enter your conflict of interest statement in the “Confidential to Editor” section, and submit your "Accept" recommendation.

Reviewer #1: All comments have been addressed

Reviewer #3: All comments have been addressed

Reviewer #4: All comments have been addressed

2. Is the manuscript technically sound, and do the data support the conclusions?

Reviewer #1: Yes

Reviewer #3: Yes

Reviewer #4: Yes

3. Has the statistical analysis been performed appropriately and rigorously? 

Reviewer #1: N/A

Reviewer #3: Yes

Reviewer #4: N/A

4. Have the authors made all data underlying the findings in their manuscript fully available?

Reviewer #1: Yes

Reviewer #3: Yes

Reviewer #4: (No Response)

5. Is the manuscript presented in an intelligible fashion and written in standard English?

Reviewer #1: Yes

Reviewer #3: No

Reviewer #4: Yes

6. Review Comments to the Author

Reviewer #1: (No Response)

Reviewer #3: Thank You for this opportunity to review this paper. I really find it interesting and informative about genital mutilation and prevailing social norms. You have really pick the important topic to discuss and present. Not much but few observation on the methodology section. Since you have mentioned this is qualitative study, it would be better if you mention what type of qualitative study it is e.g. explorative. Also, regarding the age of respondent you have mentioned 16-63 years, i am not clear on what evidence you have choosen this age range. I personally feel that while taking any age range of respondent standard age range for the study will be considered appropriate and of-course the basis of choosing only 26 respondent.

Reviewer #4: This is an excellent in-depth article on an important subject. The qualitative analysis is very well described and skillfully presented. The insights learned are valuable. My only suggestion would be to include a more specific description of the themes/findings in the results section of the abstract.

7. PLOS authors have the option to publish the peer review history of their article (what does this mean?). If published, this will include your full peer review and any attached files.

Reviewer #1: Yes: Dr Catrin Evans

Reviewer #3: No

Reviewer #4: Yes: Rebecca A Giguere

---

## [Author Response · Author response to Decision Letter 1]

9 Apr 2020

*Response to reviewers*

We would like to thank the academic editor and the reviewers for their valuable time and feedback. 

In this letter, we provide point-by-point response to the reviewers.

Response to reviewer #3

1. Thank You for this opportunity to review this paper. I really find it interesting and informative about genital mutilation and prevailing social norms. You have really pick the important topic to discuss and present. 

Response: Thank you for your positive feedback. It is highly appreciated. 

2. Not much but few observation on the methodology section. 

a. Since you have mentioned this is qualitative study, it would be better if you mention what type of qualitative study it is e.g. explorative.

Response: Done. Line 141. 

b. Also, regarding the age of respondent you have mentioned 16-63 years, i am not clear on what evidence you have choosen this age range. I personally feel that while taking any age range of respondent standard age range for the study will be considered appropriate 

Response: We agree that the standard age range for women of reproductive age-group (15-49 years) is the most appropriate age range for the study. As shown in table 1, all of our respondents, except for two key informants, are in the standard age-group. Based on the relevance of the individual experiences of these key informants to our research questions, we have decided to include them regardless of their age. 

c. The basis of choosing only 26 respondent.

Response: We have stopped recruitment once no new themes were observed. This is now clarified in line 184.

Additional comments from reviewer #3:

d. Arrange (the marital status in table 1) in ascending or descending order. 

Response: done.

e. Is the guiding question same during each time of interview? 

Response: Yes, we used a semi-structured interview guide (see S1 Appendix). 

f. Why we interview same individual twice and thrice?

Response: FGC is a personal and sensitive topic. Hence, our main aim during the first interviews was to build trust and invite the participants' own narrations of FGC related experiences (lines 190-191). In the second interviews we further encouraged the participants' own narrations of FGC related experiences by revisiting the narrations from the first interviews and probing for confirmations, corrections, and elaborations (lines 203-205). The main purpose of the third interviews was to member check our analysis by discussing our preliminary interpretations of patterns and emerging themes with the participants (lines 208-209).

g. Why there is no recording of any interview? Isn't this will generate recall bias.

Response: Only first interviews were not recorded. To minimize the risk of recall bias, MMZ recorded her recollections, as well as her interpretations of the interview immediately after each interview (lines 192-194).

Response to reviewer #4

1. This is an excellent in-depth article on an important subject. The qualitative analysis is very well described and skillfully presented. The insights learned are valuable. My only suggestion would be to include a more specific description of the themes/findings in the results section of the abstract.

Response: Thank you for your positive feedback, it is highly appreciated. A specific description of the themes is now included in the results section of the abstract (lines 31-34).

Yours sincerely 

Mai Mahgoub Ziyada

MSc, MPhil and MBBS

PhD Candidate

Section for Trauma, catastrophes and forced migration - NKVTS

Institute of health and society, Faculty of Medicine - UiO

m.m.ziyada@nkvts.no

---

## [Decision Letter · Decision Letter 2]

6 May 2020

Sexual norms and the intention to use healthcare services related to Female Genital Cutting: A qualitative study among Somali and Sudanese women in Norway.

PONE-D-19-30005R2

Dear Dr. ziyada,

We are pleased to inform you that your manuscript has been judged scientifically suitable for publication and will be formally accepted for publication once it complies with all outstanding technical requirements.

With kind regards,

Nülüfer Erbil, Ph.D, Prof.

Academic Editor

PLOS ONE

Additional Editor Comments (optional):

Reviewers' comments:

Reviewer's Responses to Questions

**Comments to the Author**

1. If the authors have adequately addressed your comments raised in a previous round of review and you feel that this manuscript is now acceptable for publication, you may indicate that here to bypass the “Comments to the Author” section, enter your conflict of interest statement in the “Confidential to Editor” section, and submit your "Accept" recommendation.

Reviewer #3: All comments have been addressed

2. Is the manuscript technically sound, and do the data support the conclusions?

Reviewer #3: Yes

3. Has the statistical analysis been performed appropriately and rigorously? 

Reviewer #3: Yes

4. Have the authors made all data underlying the findings in their manuscript fully available?

Reviewer #3: Yes

5. Is the manuscript presented in an intelligible fashion and written in standard English?

Reviewer #3: Yes

6. Review Comments to the Author

Reviewer #3: Thank You for providing me the opportunity to review your paper. All the suggestions are well revised and answered. I found methodology and findings are well presented and insightful.

7. PLOS authors have the option to publish the peer review history of their article (what does this mean?). If published, this will include your full peer review and any attached files.

Reviewer #3: Yes: Anu Bista

---

## [Editor Report · Acceptance letter]

7 May 2020

PONE-D-19-30005R2 

Sexual norms and the intention to use healthcare services related to Female Genital Cutting: A qualitative study among Somali and Sudanese women in Norway. 

Dear Dr. Ziyada:

I am pleased to inform you that your manuscript has been deemed suitable for publication in PLOS ONE. Congratulations! Your manuscript is now with our production department. 

With kind regards,

on behalf of

Mrs. Nülüfer Erbil 

Academic Editor

PLOS ONE